# Impact of Collaborative Care on Depression in Patients Aged 60+: A Secondary Analysis of the GermanIMPACT Study on Behavioural Activation

**DOI:** 10.3390/bs15040462

**Published:** 2025-04-03

**Authors:** Sigrid Boczor, Sanaz Ashrafi, Frederike Bjerregaard, Christiane Bleich, Thomas Grochtdreis, Dagmar Lühmann, Martin Härter, Lars Hölzel, Michael Hüll, Iris Tinsel, Martin Scherer, Thomas Kloppe

**Affiliations:** 1Department of General Practice and Primary Care, University Medical Center Hamburg-Eppendorf, 20246 Hamburg, Germany; 2School of Medicine, University of Galway, University Road, H91 TK33 Galway, Ireland; 3Department of Medical Psychology, University Medical Center Hamburg-Eppendorf, 20246 Hamburg, Germany; 4Department of Health Economics and Health Services Research, Hamburg Center for Health Economics, University Medical Center Hamburg-Eppendorf, Martinistraße 52, 20246 Hamburg, Germany; 5Health Services Research, Oberberg Group, 10117 Berlin, Germany; 6Department of Psychiatry and Psychotherapy, University Medical Center Mainz, 55131 Mainz, Germany; 7Department of Psychiatry and Psychotherapy, Medical Center—University of Freiburg, Faculty of Medicine, University of Freiburg, 79106 Freiburg, Germany; 8Emmendingen Center for Psychiatry, Clinic for Geriatric Psychiatry and Psychotherapy, University of Freiburg, 79312 Emmendingen, Germany; 9Institute of Medical Biometry and Statistics, Section of Health Care Research and Rehabilitation Research, Faculty of Medicine and Medical Center, University of Freiburg, 79106 Freiburg, Germany

**Keywords:** depression, primary care, collaborative care, behavioural activation, old age, late life, nurse-led intervention

## Abstract

Depressive disorders are highly prevalent among older adults (60+) in Europe. Activating these patients was a core component of the GermanIMPACT study, which evaluated collaborative care in a cluster-randomized primary care setting. The intervention group showed a significant improvement in PHQ-9 remission. The aim of this secondary analysis was to investigate which activities were planned and whether their implementation or non-implementation was associated with depressive symptoms (PHQ-9) after 12 months. Behavioural activation data were collected by the care managers. A categorization for activity type (collected as free text) and activity implementation status was developed. The association of successfully implemented activities, planned-not-implemented activities, and the number of activities per patient with the 12-month PHQ-9 total score was calculated using logistic regressions (adjusted for age, gender, living situation/baseline PHQ-9). A total 2188 activities were planned for 136 patients; 66% were successfully implemented. Mean age was 71 (±7) years (78% female; 52% living alone). Activities focusing on “self-care/spirituality” improved the PHQ-9 outcome (OR 1.540; *p* = 0.048), while planned-not-implemented activities overall worsened it (OR 1.16; *p* = 0.007). Patient activation is key to treating depressive symptoms in old age. Particularly ‘self-care/spirituality’ activities could be planned, and organizational activities should be closely supported.

## 1. Introduction

Depression is one of the most common mental disorders in older people and the annual prevalence of depression in patients over the age of 65 is 14% ([3]). In primary care, approximately 10% of this age group suffer from depression ([27]). In addition to the increased risk of physical illness, functional impairment and dementia in depressed older patients ([15]), affected patients also have a comparatively higher use of the healthcare system ([7]). Furthermore, the demographic change is highlighting the need for efficient and cost-effective treatments for depressive symptoms in old age, with this patient group posing a particular challenge for care provision ([2]). Adequate treatment requires interventions that are tailored to the needs of this patient group.

This study takes a detailed look at behavioural activation, one of the evidence-based non-pharmacological therapies in the treatment of depression in older people ([1]). The behavioural model of depression, introduced by Peter M. Lewinsohn in 1974, posits that depression arises from low positive reinforcement and exposure to negative events. This results in reduced engagement in enjoyable activities, reinforcing depressive moods ([24]). Behavioural activation, a key element of cognitive behavioural therapy (CBT) and applied stand-alone is as effective in treating depression as CBT ([9]) and can be implemented by less specialized professionals, including psychiatric nurses ([13]; [28]). It aims to increase rewarding activities in patients’ lives, thereby breaking the “vicious cycle of depression” and improving symptoms.

This paper is based on data from the study ‘GermanIMPACT—coordinated treatment of age-related depression in primary care’ ([19]). The intervention aimed to help patients achieve a long-term improvement in mood, and thus a reduction in their depressive symptoms, through a guided program of activities ([14]). In the nurse-led intervention, care managers provided patients aged 60+ and suffering from moderate depressive symptoms with an easily accessible telephone-based support as part of a stepped care model based on severity and treatment response. The control group received treatment as usual from general practitioners (TAU). Conducted over the course of one year, the complex intervention proved more effective than TAU; 25.6% of the participants (*n* = 139) in the intervention group experienced a remission of depressive symptoms compared to 10.9% in TAU (*n* = 109) ([19]).

Behavioural activation was the main component of the intervention alongside monitoring, psychoeducation, relapse prophylaxis, and training in problem-solving if indicated. Care management was typically provided via telephone in two weekly treatment sessions. Care managers discussed activities with the patients, reviewed tasks for behavioural activation, analysed potential obstacles to carrying out the activities, and planned new activities.

The aim of this study is to analyse the planned and implemented activities as part of behavioural activation in the intervention group of the GermanIMPACT study to determine which activities can be used most effectively for behavioural activation in older people. In addition, the study examined which activities were planned and whether their implementation or non-implementation showed an association with depressive symptoms, as measured by the Patient Health Questionnaire-9 (PHQ-9), after 12 months. The research also aims to identify the number of activities per patient, along with any problems and barriers to their implementation.

## 2. Materials and Methods

### 2.1. Study Sample

The methods of the study on the collaborative treatment of patients with late-life depression in German primary care (GermanIMPACT), including the details of the sequence of intervention sessions, have been described in detail elsewhere ([19]). Briefly, the main GermanIMPACT study was a 12-month bi-centric, cluster-randomized, controlled trial aimed to determine the efficacy of the IMPACT program including a specially trained care manager compared to standard primary care treatment in elderly patients with moderate depressive symptoms in Germany.

In the main GermanIMPACT study, a total of 248 patients with an overall score of 10–14 points (moderate depressive symptoms) on the PHQ-9 ([22]) and an age of at least 60 years were recruited for the study between 2012 and 2014. The primary care practices were cluster-randomized, resulting in 139 study patients in the intervention group and 109 study patients in the treatment as usual (TAU) group. TAU patients received regular primary care without any specific requirements imposed by the study protocol.

In the intervention group, after an initial face-to-face and a first telephone session with the patient the stepped care algorithm included follow-up telephone sessions (every 2 weeks) and evaluation sessions (every 8 weeks). At this meeting, it was decided whether the continued telephone sessions should take place at intervals of 4 or 2 weeks, or if a weekly problem-solving training with a maximum of 6 sessions should be carried out. The primary study endpoint was the change in depressive symptoms measured with the PHQ-9 at twelve months. Each patient provided written consent to participate prior to the onset of the study.

This study included patients in the intervention group, with care manager documentation available for only 136 of the 139 patients. The control group of the GermanIMPACT main study with 109 patients did not receive activity planning and was not monitored. Therefore, the control group could not be included in this analysis.

### 2.2. Measures

In this secondary analysis, the information on behavioural activation in the intervention group (monitoring; psychoeducation; activity development; if necessary: medication advice and/or problem-solving training) recorded by the care manager in structured log sheets and entered into a database, was evaluated. The patients chose the activities at the beginning of each telephone-based treatment session according to their own ideas or using a list according to [24] ([24]), which was modified for the study ([24]). A corresponding category system for activity type (physical, daily duties, social, IMPACT-related, at rest, organizational, excursions and travel, self-care and spiritual, health care, culture and education, other) as well as for activity implementation (activity as planned, i.e., successfully implemented; partially implemented; activity replaced; implementation unknown; not implemented) was developed from the free-text information. The total number of planned activities was reported. Further, the frequency of success of the implementation of activities was calculated and reported for the categories successful, planned-not-implemented, and the rest of activities, i.e., the partially implemented, replaced, and those with unknown implementation status were summarized in one category. Accordingly, the total number of activities per patient was calculated in the categories successful and planned-not-implemented activities, and the mean, standard deviation of the mean, and the maximum are reported.

The PHQ-9 total score has a scale of 0–27 point with a Cronbach’s α of 0.89 for patients in primary care ([22]). Its interpretation states that the higher the score, the more severe a patient’s depressive symptoms are, with 10–14 points indicating moderate depression ([22]).

### 2.3. Statistical Analyses

Absolute and relative frequencies are given for patient sex, age, living situation, level of education, employment, the patient’s perception of the sufficiency of their financial resources, the physical activity categories, and activity implementation status. A possible association of successfully implemented activities, as well as of the number of activities per patient, with a PHQ-9 total score moving towards improvement after 12 months (at least +1 point or more versus no change or deterioration) was calculated using logistic regressions. Additionally, a possible association with a decrease of 1 point or more, compared to no change or an increase in PHQ-9 total score at 12 months, was examined regarding planned-not-implemented activities. Univariate and multivariate logistic regression models were adjusted for age, gender, living situation, and the baseline PHQ-9 total score ([18]). The beta coefficient and standard error, the Wald statistic and *p*-value, and the odds ratio (OR) with 95% confidence intervals were presented. Additionally, given quality parameters of the logistic regression models were χ^2^, *p*-value of omnibus tests of model coefficient, explained variance, correct classification performance, sensitivity, and specificity. Univariate regressions were only reported for statistically significant results. Statistical analyses were performed using IBM SPSS for windows version 29.0.

## 3. Results

The mean age of the patients with documented activities (*n* = 136 of 139 patients in the GermanIMPACT intervention group) was 71 (±7) years; 78% of participants were female, 52% lived alone (Table 1).

In total, 2188 planned activities had been documented in the 136 patients, including 573 cases of activities not performed, which were summarized in eleven categories (Table 2; see Table 3).

In total, 66% of the planned activities were successfully implemented (Table 3).

Twenty-three patients (16.5%) showed no change in the PHQ-9 total score (±0 points) after 12 months.

### 3.1. Improvement in the PHQ-9 Total Score

Almost two-thirds (61.8%) of the 136 patients showed an improvement of at least 1 point in the PHQ-9 total score after 12 months. In the adjusted univariate logistic regression the following four activity categories, when successfully implemented, showed an association with an improved PHQ-9 total score after 12 months: self-care and spirituality [*p* = 0.010; OR 1.639 (1.126–2.387)]; daily duties [*p* = 0.032; OR 1.032 (1.023–1.649)]; physical activity [*p* = 0.030; OR 1.226 (1.020–1.474)]; excursions and travel [*p* = 0.028; OR 1.378 (1.035–1.834)].

The adjusted multivariate model showed an association of the baseline PHQ-9 total score [*p* = 0.001; OR 1.173 (1.065–1.293)] and “self-care and spirituality” activities [*p* = 0.048; OR 1.540 (1.005–2.360)] with improved PHQ-9 total scores after 12 months (Table 4).

The mean ± SD total number of successfully implemented activities in the female and male patients was similar, with a sum of 10.75 (±9.14 (maximum 41) and 10.37 (±9.18, (maximum 29), ranging, however, from 0 to >20 activities per patient. The adjusted improvement in the PHQ-9 total score after 12 months showed a significant association with the total number of activities performed per patient [*p* = 0.005, OR 1.071 (1.021–1.124); Table 5].

### 3.2. Deterioration of the PHQ Total Score After 12 Months

The successfully implemented, i.e., carried-out activity category “organizational” showed an association on the deterioration of the PHQ-9 total score in the adjusted univariate logistic regressions [*p* = 0.035; OR 1.378 (1.024–1.856)].

Regarding the planned-not-implemented activities, the activity category “health care” showed in univariate logistic regression an association with a deteriorated PHQ-9 total score after 12 months [*p* = 0.039; OR 1.632 (1.024–2.600)] as well as the category “IMPACT-related” [*p* = 0.033; OR 1.547 (1.036–2.310)]. In the adjusted multivariate logistic regression model regarding the non-implemented activities, an association with a deteriorated PHQ-9 total score after 12 months was shown by the baseline PHQ-9 total score [*p* = 0.003; OR 0.833 (0.738–0.941)] and the IMPACT-related activities [*p* = 0.029; OR 1.679 (1.055–2.673); Table 6].

Also, the total sum of non-implemented activities showed an association with a deteriorated PHQ-9 total score after 12 months [B = 0.148; SE = 0.055; Wald = 7.322; *p* = 0.007; OR 1.160 (1.042–1.291); adjusted for patient sex, age, living situation and baseline PHQ-9; χ^2^(14) = 16.914; *p* = 0.005. Explained variance = 18.1%. Correct classification performance = 80.9%. Sensitivity = 13.8%. Specificity = 99.1%. *N* = 136 patients].

## 4. Discussion

This study analysed the planned and implemented activities as part of behavioural activation in the intervention group of the GermanIMPACT study, aiming to identify which activities were the most effective for behavioural activation in older adults with moderate depressive symptoms in a primary care setting. The study further investigated whether the planning and implementation or non-implementation of these activities were associated with depressive symptoms, as measured by the PHQ-9, after 12 months.

In the GermanIMPACT study, older GP patients treated for depression engaged in a wide range of social and individual activities during treatment sessions. Overall, a pool of 2188 planned activities resulted in an implementation rate of 66%. Around a quarter (*n* = 573) of the total 2188 planned activities could not be successfully implemented.

An association was found between the successful implementation of the activities and the improvement in depressive symptoms. The stronger the depressive symptoms of a patient were at baseline, the more the PHQ-9 total score improved after 12 months. These patients had a greater potential for improvement than patients with a better baseline score. In addition to the number of activities carried out per patient as part of the intervention and the baseline PHQ-9 total score, we also identified specific activity categories that showed a significant association with the improvement in depressive symptoms and could be differentiated from other categories. Particularly noteworthy here are activities in the “self-care and spiritual” category, which showed a significant association with the improvement in depressive symptoms. Furthermore, compulsory activities, physical activities and excursions, and travelling showed a significant positive association with the improvement in depressive symptoms.

A significant deterioration in depressive symptoms was only observed in the univariate analysis for organizational activities. This could reflect the results of the qualitative study by [21] ([21]). They examined problem-centred interviews with 26 patients from the intervention group from the GermanIMPACT study and identified five distinct types of utilization regarding pleasant activations, influenced by the patients’ initial and ongoing activity levels during therapy: “activatable relief seekers”, “active relief seekers”, “active relaxation seekers”, “passive problem-solving seekers”, and “passive relief seekers”. Although these types exhibited highly varied levels of activation, no significant association with overall effectiveness of the intervention was found. Instead, patient-specific characteristics within these response types seemed to play a more significant role in the effectiveness of the GermanIMPACT intervention. Factors such as the presence of a recent depression trigger, moderate symptom burden, good mobility, limited comorbidity, general interest in activities, a solid social network, and a healthy self-esteem appeared to influence treatment outcomes more than the activities themselves ([21]).

Physical activity emerged as the most frequently planned category with a high planning and implementation rate of 71% and significantly improved PHQ-9 total scores in univariate analyses, aligning with the American IMPACT study ([29]). The relevance of physical activity for mental health in older adults is crucial in behavioural activation ([8]; [29]) and well-documented in the literature ([35]). Older adults often prefer physical activity as a treatment option for depression, alongside medication and psychotherapy ([25]), if comorbidities that limit their mobility are not an obstacle. This main obstacle is also described in two further interview studies with care managers and GPs that were conducted for the qualitative process evaluation of GermanIMPACT ([5]; [6]). The adaptation of the intervention for people with limited mobility, e.g., with chronic pain, could be promoted ([12]).

However, the patients of the GermanIMPACT study did not implement one-third of all activities. As reported by [4] ([4]), they stated, in descending order, a lack of time, the influence of depressive symptoms, and existing acute health problems as the main problems and obstacles that contribute to the failure of activation ([4]). These findings are consistent with other studies ([25]; [36]; [37]). However, reasons for non-implementation reported by patients may differ from the underlying factors. For example, [25] ([25]) found that indecision in treatment preferences is common in depressed older adults. The often reported “lack of time” may also mask deeper issues such as depression ([25]). Considering the subjective perception of physical health in treatment strategies can significantly enhance therapeutic outcomes ([10]).

“Daily duties”, such as cooking meals, doing the laundry, or cleaning, were often completed successfully and linked to improved PHQ-9 total scores, possibly due to added structure and routine, which can enhance mental health. This approach to treatment has received too little attention to date, although its effectiveness is well known ([26]). Even the study by [21] ([21]) showed that the successful implementation of non-pleasurable activities within GermanIMPACT has a mood-enhancing and self-efficacy-strengthening effect. In future, the intervention could utilize this aspect of self-management support in a more effective way, particularly to support chronically depressed patients ([36]).

The activity category “excursions and travel” showed positive effects in univariate analysis. They had a high non-completion rate due to logistical and financial challenges, and external influences such as poor weather conditions or unavailable accompanying persons. Planning alternatives for such activities is essential. The general implementation rate might be boosted by incorporating the concept of expectation violation into behavioural activation when treating depression. Key to this is the realization that depressed patients often discover that carrying out planned activities is less tiring and more enjoyable than they had anticipated ([23]; [30]).

When patients failed to implement their planned activities in the “health care” and “IMPACT-related” categories, univariate logistic regression showed an association with a worsened PHQ-9 total score at 12 months. This association is easy to explain but even the “planned-successfully-implemented” category “organizational tasks”, like organizing the sale of an apartment or applying for a disabled person’s pass, was significantly associated with negative effects on depressive symptoms in univariate analysis. This minor observation may sensitize us that coping strategies such as mindfulness can mitigate such stressful tasks. Incorporating self-care activities alongside necessary tasks could probably buffer these stress effects ([11]). These findings were consistent with our regression analyses that showed significant differences between activity categories. The activity category “self-care and spirituality” stood out as positively impacting PHQ-9 total scores, aligning with studies linking spiritual well-being to less depression and stronger resilience ([31]; [38]). Self-care practices, including healthy lifestyle habits, such as stress management techniques like yoga and mindfulness, as well as maintaining good sleep hygiene, lead to a reduction in PHQ-9 total scores even in multimorbid elderly patients ([20]).

Although social activities had high planning and implementation rates in our study, they did not show a statistically significant impact on depression improvement as assessed with the PHQ-9 in regression analyses. The lack of a significant effect of social activities on a change in the PHQ-9 total score towards an improvement in our study might be due to insufficient emphasis on interactions that involve personal engagement. A distinction between passive and active social engagement, or an investigation into whether the quality of interactions influences the results, provides more differentiated results. In the qualitative sub-study by [21] ([21]), we saw that responders mainly reported a solid social network with communicative exchange and support. Non-responders were not in close relationships, and even when they tried to make personal contacts, they often failed to overcome their loneliness.

Despite the statistical non-significance in our data, the relevance of social interactions is supported by other studies ([32]; [33]). Regular social engagement can decrease depression risk by up to 13% in older adults ([31]), particularly those involving close personal interactions ([33]). Future research and clinical practice could emphasize the effects of social interaction and encourage activities focused on such engagement.

### 4.1. Strengths and Limitation

This analysis were based on the 2188 planned activities of patients in the GermanIMPACT study ([19]) and represents the most comprehensive evaluation of the interventions to date (97% = 136 of 139 patients) compared to another study of [29] ([29]) with data from only two-thirds of their patients ([29]). Also new in research is the investigation of the reasons for the failure of behavioural activation.

However, the current analysis only presented results of the intervention group of the main GermanIMPACT study, as the control group did not include any activity assessment. Nevertheless, since we assume that the main symptom of depression is lack of motivation, we also assume that without targeted activity planning only a few activities took place in the equally depressed control group.

Since the documented activities were recorded by specifically trained care managers, it cannot be completely ruled out that them may not accurately reflect the patients’ actual descriptions. To keep the reporting bias as low as possible, there was a supervision meeting every 4 weeks with all four care managers and the study coordinator to ensure quality. During these meetings, the documentation was standardized, for instance. Nevertheless, the category system was self-developed, and an external validation has not been yet carried out.

However, we also conducted a structured supervised intervention with stringent quality control measures in place, which is essential for maintaining a very low level of variance even in documentation. Patients may have had difficulty explicitly stating reasons and putting their thoughts into words. Admitting that activities were not carried out and the exact reason for this may have been uncomfortable for patients. Nevertheless, a better data specification is hardly possible in this context. Even if patients classify their activities themselves based on predefined lists, a different depiction would be possible for various reasons, such as social desirability.

The study is strongly focused on the planning of pleasant activities, but the planning of activities that are not necessarily described as mood-enhancing was also proven to be relevant. These activities must also be carried out in old age and are part of the lives of older patients. However, the negative influence of organizational activities, for example, on the PHQ-9 total score suggests that support services for this type of activity could help those affected.

Due to the character of a secondary analysis, all results should be interpreted with caution and cannot be interpreted causally. Consequently, our results should rather be understood as indicators that would need to be verified in a confirmatory randomized controlled trial.

### 4.2. Implications

This study provides valuable insights to strengthen the effectiveness of future care approaches for patients aged 60+ with moderate depressive symptoms. Based on implementation rates, correlation and regression analyses, as well as clinical relevance, the following categories of activities could be considered, depending on the patient’s individual possibilities:Self-care and spirituality: individually plannedDaily duties: maintain a personal daily routinePhysical activity: individually plannedSocial activities: consider personal interaction with othersExcursions and travel: consider planning alternatives

Considering these recommendations, the exemplary list of activities used in the study could be adapted and prioritized ([24]). Activities from the categories mentioned above could be recommended and planned more frequently for patients. In addition, an important factor for a decrease in depressive symptoms is probably the number and heterogeneity of activities per patient. Ideally, activities belonging to these categories could be performed frequently, with the aim of integrating several categories into the activation process instead of repeatedly performing activities from a single category.

The results of this research could serve as a focusing guide for health care professionals who wants to use behavioural activation as a treatment for older patients with depressive symptoms. Additionally, this focus on specific activities may also be useful for the design of further studies on collaborative care for patients with depression. As we have seen that mobility limitations are a major barrier to the implementation of the activities, future approaches should pursue personalized planning strategies for patients with limited mobility.

The utilization of care managers in routine care would be particularly beneficial for older patients with depression, especially regarding organizational demands. This approach could help to avoid negative impacts on depressive symptoms and enhance positive influences through targeted activity planning like a “safety net” ([34]). Patients with very mild symptoms could be chosen for targeted activity planning with lower priority, as the higher the initial PHQ-9 total score, the greater the improvement after 12 months, given there is more room for progress. Both coordinated intervention models and the presence of care managers in the GP care of depressed patients were rated positively by GPs in GermanIMPACT ([5]). They could help to improve the quality of care for people suffering from depression and relieve the burden on GPs ([17]). In addition, care managers could represent a low-threshold option for patients with depression, as it is difficult for older patients to access psychotherapy and they are not necessarily willing to attend ([21]). Therefore, the methods of the collaborative treatment from GermanIMPACT have the potential to support the comprehensive integration of mental health into primary care, as advocated by the World Organization of National Colleges, Academies and Academic Associations of GPs, and the WHO ([16]).

## Figures and Tables

**Table 1 behavsci-15-00462-t001:** Sociodemographic of the GermanIMPACT study patients with documented activities in the intervention group (*N* = 136).

Category	Intervention Group, *N* = 136 Patients
**Gender**	
Female, n (%)	106 (78%)
Male, n (%)	30 (22%)
Mean age, (range)	71 years (52–88)
Living alone, (%)	51.5%
**Level of education, (%)**	
Completed university degree	12.5%
University entrance qualification	9.6%
Secondary school certificate	51.5%
Intermediate school certificate	22.8%
Other or no educational qualification	3.6%
**Employment, (%)**	
Still employed	10.3%
Retired	65.4%
Housewife/househusband	14.0%
Unemployed	3.7%
Other or missing	6.6%
**Sufficient financial resources (Subjective patient assessment), (%)**	
Yes	48.9%
Partially	39.3%
No	11.9%

**Table 2 behavsci-15-00462-t002:** Activities planned by the patients in the GermanIMPACT intervention group ^1^ ordered by relative frequency.

Activity Category	Examples	Relative Frequency	N
Physical	“Go to the gym twice a week”	15.9%	347
Daily duties	“Cook food” “Do the laundry”	13.3%	290
Social	“Visiting relatives”, “Activity at the senior citizens’ meeting place”	12.4%	272
IMPACT related	“Fill out daily logs” “View activity list”	10.8%	237
At rest	“Listen to the radio” “Read a book”	10.3%	225
Organizational	“Plan the sale of your apartment” “Organize household help”	9.5%	207
Excursions and travel	“Holiday in ‘X’ with husband” “Trip by bus”	8.5%	186
Self-care and spiritual	“Meditate daily” “Go to church on Christmas Eve”	8.4%	183
Health care	“Make an appointment with an orthopaedist” “Get a prescription from your GP”	5.2%	113
Culture and education	“Attend a lecture on topic ‘X’” “Go to the theatre”	3.4%	75
Other	“Wants to cancel planned kale dinner” “Cemetery visits: ‘X’th anniversary of the husband’s death”	2.4%	53

^1^ *N* = 136 patients. GP = General practitioner, X = anonymized.

**Table 3 behavsci-15-00462-t003:** Implementation of planned activities ^1^ ordered by relative frequency of success.

Activity Category	Successful	Not Implemented	Partial/Unknown/Activity Replaced
Daily duties	76.9% (223)	15.5% (45)	7.6% (22)
At rest	73.3% (165)	20.0% (45)	6.7% (15)
Social	71.0% (193)	24.3% (66)	4.8% (13)
Culture and education	70.7% (53)	21.3% (16)	8.0% (6)
Physical	70.6% (245)	18.7% (65)	10.6% (37)
Self-care and spiritual	69.4% (127)	21.9% (40)	8.7% (16)
Other	67.9% (36)	28.3% (15)	3.8% (2)
Excursions and travel	65.1% (121)	30.6% (57)	4.3% (8)
Organizational	52.7% (109)	40.1% (83)	6.2% (15)
Health care	51.3% (58)	37.2% (42)	11.6% (13)
IMPACT related	47.7% (113)	41.8% (99)	10.6% (25)
Total	66.0% (1443)	26.2% (573)	7.8% (172)

^1^ *N* = 2188 planned activities in total.

**Table 4 behavsci-15-00462-t004:** Multivariate logistic regression model with the target parameter improvement in the PHQ-9 after 12 months.

Activity Category ^1^	B	SE	Wald	*p*-Value	Odds Ratio (95% CI)
IMPACT-related	0.278	0.295	0.887	0.346	1.321 (0.740–2.355)
Social	−0.102	0.096	1.137	0.286	0.903 (0.748–1.089)
At rest	0.106	0.135	0.617	0.432	1.122 (0.854–1.447)
Self-care and spiritual	0.432	0.218	3.925	0.048	1.540 (1.005–2.360)
Daily duties	0.207	0.141	2.151	0.143	1.230 (0.933–1.622)
Organizational tasks	−0.196	0.160	1.503	0.220	0.822 (0.600–1.125)
Physical	0.057	0.100	0.319	0.572	1.058 (0.869–1.288)
Health care	−0.167	0.173	0.932	0.334	0.846 (0.602–1.188)
Culture and education	−0.028	0.177	0.025	0.875	0.973 (0.688–1.376)
Excursions and travel	0.223	0.176	1.610	0.205	1.250 (0.885–1.765)
Age	−0.018	0.031	0.353	0.553	0.982 (0.925–1.043)
Gender	0.073	0.523	0.020	0.889	1.076 (0.386–2.996)
Living situation	0.074	0.453	0.026	0.871	1.076 (0.443–2.616)
Baseline PHQ-9	0.160	0.049	10.491	0.001	1.173 (1.065–1.293)
(Constant)	−0.636	2.312	0.076	0.783	0.530

^1^ The activity category “Other” was not included in the model due to the non-specific activities. χ^2^(14) = 37.373; *p* = 0.001. Explained variance = 32.7%. Correct classification performance = 71.3%. Sensitivity = 81.0%. Specificity = 55.8%. *N* = 136 patients.

**Table 5 behavsci-15-00462-t005:** The sum of performed activities per patient in the adjusted logistic regression model with the target parameter being the improvement in the PHQ-9 after 12 months.

Parameter	B	SE	Wald	*p*-Value	Odds Ratio (95% CI)
Sum of performed activities per patient	0.069	0.025	7.891	0.005	1.071 (1.021–1.124)
Gender	−0.054	0.486	0.012	0.912	0.947 (0.365–2.457)
Age	−0.031	0.028	1.163	0.281	0.970 (0.917–1.025)
Living situation	0.024	0.404	0.004	0.952	1.025 (0.464–2.262)
Baseline PHQ-9	0.152	0.044	11.759	0.001	1.165 (1.067–1.271)
(Constant)	0.451	2.120	0.045	0.831	1.571

χ^2^(5) = 24.352; *p* < 0.001. Explained variance = 22.3%. Correct classification performance = 69.1%. Sensitivity = 82.1%. Specificity = 48.1%. *N* = 136 patients.

**Table 6 behavsci-15-00462-t006:** Multivariate logistic regression model with the target parameter being the deterioration of the PHQ-9 total score after 12 months regarding planned-not-implemented activities.

Activity Category ^1^	B	SE	Wald	*p*-Value	Odds Ratio (95% CI)
IMPACT-related	0.518	0.237	4.771	0.029	1.679 (1.055–2.673)
Social	0.409	0.220	3.459	0.063	1.506 (0.978–2.318)
At rest	0.114	0.276	0.171	0.679	1.121 (0.653–1.924)
Self-care and spiritual	0.202	0.314	0.414	0.520	1.224 (0.661–2.265)
Daily duties	0.409	0.314	2.437	0.118	1.632 (0.882–3.020)
Organizational tasks	−0.047	0.167	0.079	0.778	0.954 (0.687–1.324)
Physical	0.333	0.308	1.170	0.279	1.395 (0.763–2.550)
Health care	0.408	0.245	2.762	0.097	1.504 (0.929–2.433)
Culture and education	0.532	0.567	0.880	0.348	1.702 (0.560–5.168)
Excursions and travel	−0.869	0.451	3.709	0.054	0.419 (0.173–1.016)
Age	0.035	0.038	0.830	0.362	1.036 (0.961–1.117)
Gender	0.509	0.626	0.663	0.416	1.664 (0.488–5.670)
Living situation	0.798	0.568	1.973	0.160	2.221 (0.729–6.764)
Baseline PHQ-9 total score	−0.182	0.062	8.659	0.003	0.833 (0.738–0.941)
(Constant)	−3.644	2.950	1.525	0.217	0.026

^1^ The activity category “Other” was not included in the model due to the non-specific activities. χ^2^(14) = 28.587; *p* = 0.012. Explained variance = 29.4%. Correct classification performance = 81.6%. Sensitivity = 27.6%. Specificity = 96.3%. *N* = 136 patients.

## Data Availability

The data presented in this study are available upon request from the corresponding author. The data are not publicly available due to restrictions, e.g., the data may contain information that could compromise the privacy of research participants.

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
