# Peer review of "Impact of Collaborative Care on Depression in Patients Aged 60+: A Secondary Analysis of the GermanIMPACT Study on Behavioural Activation"

_behavsci, 2025, doi:10.3390/bs15040462_

Round 1

Reviewer 1 Report

Comments and Suggestions for Authors

This study offers a comprehensive and insightful analysis of the GermanIMPACT intervention for older adults with moderate depressive symptoms. One of its key strengths lies in the large sample size (2,188 planned activities from 136 patients), which enhances the robustness of the findings. Furthermore, the detailed categorization of activities and the use of multilevel statistical analyses provide valuable insights into how specific activity types influence depressive symptoms over 12 months. The study’s focus on individualized intervention—particularly in categories such as self-care, spirituality, and physical activities—underscores its practical relevance for tailoring treatment to patient needs. Additionally, the inclusion of care managers as part of the intervention highlights an innovative, collaborative care model that could ease the burden on general practitioners (GPs) and improve patient outcomes.

Although the study acknowledges that data were recorded by trained care managers, it would be helpful to elaborate on how potential reporting bias was mitigated. Incorporating patient self-reports alongside care manager documentation could enhance the accuracy and reliability of the activity records. Given that one-third of planned activities were not implemented, a deeper exploration of the underlying causes (beyond logistical and health-related barriers) would be valuable. Future studies might consider qualitative interviews to capture psychological factors, such as motivation and perceived benefits.

Despite providing strong correlations, the study would benefit from discussing the limitations of drawing causal inferences due to its observational design. A suggestion for future research could be the use of randomized controlled trials (RCTs) to establish clearer cause-effect relationships. The finding that social activities did not significantly impact depressive symptoms warrants further discussion. Differentiating between passive and active social engagement or exploring whether the quality of interactions influences outcomes could offer more nuanced insights. Since the study suggests focusing on specific categories (self-care, physical activity, etc.), it would also be useful to propose personalized planning strategies for patients with limited mobility or organizational burdens, as these activities were linked to negative outcomes.

Overall, this study makes a significant contribution to the field of geriatric mental health by offering practical, evidence-based insights. Addressing these minor points could further refine the findings and enhance their applicability in clinical settings.

Reviewer 2 Report

Comments and Suggestions for Authors

The study addresses a highly relevant and significant topic for understanding depressive symptomatology and quality of life in individuals over 60 years old.

The introduction appropriately contextualizes the importance of the study. However, the methodology section requires greater clarity in certain aspects:

  • The type of study conducted should be explicitly stated in the methodology section and not only in the abstract.
  • The author of the PHQ-9 scale used should be specified, as well as its psychometric properties for the studied population.
  • It is necessary to clarify which author was referenced for the categorization of the free-text data collected, where the planned activities were described.
  • In line 97, it is mentioned that 248 participants were recruited between 2012 and 2014. However, in line 148, it is stated that the intervention group included 136 participants. The criteria used to define the sample should be explained.
  • Given the existence of a control group and an intervention group, it would be relevant to present the differences between them. The article only provides results for the intervention group, making it impossible to compare the groups.

The presentation of results should also be improved, particularly in the following aspects:

  • The possible response options regarding the participants' sociodemographic characterization should be specified.
  • Information on the performance of the control group should be included to allow for a more comprehensive analysis of the intervention’s effects.
